# OpenReview forum: "Prover-Verifier Games improve legibility of LLM outputs"
_ICLR.cc/2025/Conference — Submitted to ICLR 2025_

### Official Review · Reviewer_5yX2 · 2024-10-31

**Soundness:** 2
**Presentation:** 3
**Contribution:** 2
**Rating:** 3
**Confidence:** 4

**Summary:**

This paper tackles the problem of making LLMs generate reasoning output that is clear and easy to check (legibility), and proposes a training algorithm that iteratively trains a prover model (that generates reasoning output) and verifier model (that verifies whether a reasoning output is correct) to produce  (1) more legible solutions, (2) robust verifiers, and (3) 'sneaky' provers that can generate subtle flaws in reasoning.

**Strengths:**

- The paper raises an interesting question of whether there is a trade-off between performance and legibility of LLM output.
- It also proposes a conceptually simple actor-critic approach of training an LLM for legibility, by iteratively training the prover model and small verifier model, the latter providing a proxy for time-constrained human for evaluating legibility.

**Weaknesses:**

- The comparisons to related works are relatively cursory. It would be beneficial to provide more detailed elaboration on the differences between other related works and technical contribution of this work, such as:

	○ actor-critic frameworks on reasoning and planning tasks that also involve training of both models,

	○ how the notion of legibility is significantly different from other RLHF works where legibility would be a key implicit factor for human preference already by default,

	○ More detailed and technical elaboration on how this work relates to the larger body of work on explainability, beyond what the authors have described as "allowing legibility to emerge naturally", among others

- The timed human evaluation trials have a very short time-window evaluation of 45 seconds, with trial length of up to 4 hours. Details of the evaluation would be important in assessing its accuracy, especially as the authors pointed out in the main paper, the study may have potential design flaws -- information from the appendix and additional details not included should be shifted to the main paper if space permits.

	○ For example, the 45s limit significantly disadvantages longer responses even if they may be clearer or more understandable to humans -- an ablation controlling for length would be very useful.

	○ Also, details on whether there are any systemic trends of human evaluation over the trial time, similarity of questions provided during the trial, and distribution of questions shown during the trial, would be useful even if only in the appendix.

- The empirical evaluations are done only on 1 dataset. Especially for RL studies such as this, it would be important to assess whether the results are due to extensive tuning/fitting to this dataset or whether the method can be extended to other datasets

	○ This should include out-of-distribution results especially for claims regarding the better performance and robustness in verifiers, and legibility of provers, in case the main results of this results are specifically overfitted to just this dataset.

- The notion of legibility is vaguely defined, based primarily on the time-constrained human evaluator trials that the authors pointed out may have design flaws. A more careful design of these trials would significantly improve the paper.

	○ For example, there could be further information on why the humans may find one solution clearer than the other. This would provide more details on the characteristics that are most influential, (e.g., 'I just don't have enough time to read it', ' the answers are more spaced out' etc) which may not even require RL training to implement in the future for better legibility.

**Questions:**

Please refer to the weaknesses. Clarifications and responses to each of them would help, especially with regards to the technical contributions, lack of empirical results on OOD and other datasets which is a major flaw, clarity on definition of legibility, and potential design flaws of the human trials which are critical for the main claims of the paper.

---

> ### Author Response · Authors · 2024-11-17
>
> We thank the reviewer for recognizing we study an interesting tradeoff between performance and legibility, as well as proposing a clear method.
>
> Related works: We updated the related works section on actor-critic, RLHF, and explainability literature. We thank the reviewer for bringing up those works. Our work is complementary to RLHF. Obtaining human labels for legibility is expensive (requiring pairs of comparison data). Our main motivations for exploring alternatives to RLHF are in Section 1 (Line 42-45), i.e., using small LLM as proxy is more scalable, allows for continuous parameterization, and can offer incremental signals when the tasks are too difficult.
>
> Details of human study are in Appendix K; we did not include it in the main text due to space limit. To recap:
>
> 1. Short time limit: time-limited human judge accuracy is the main measure of human legibility in our paper. It’s one of many possible measures, and we welcome better suggestions. We used a 1 min limit in the pilot campaign and the tasks turned out too easy.
>
> 2. Control for solution lengths: Legibility has multiple dimensions: brevity, good format, simple vocabulary, etc. Length is one of the variables. We did not single out any single component of legibility but rather measure it holistically via time-constrained human judge accuracy. Our goal is to be legible in the time frame that humans have to spend on the task. If we allow infinite time, humans will do the task themselves. We want AIs to be helpful within the time constraints that humans have (though in this case they’re imposed by experimenters).
>
> 3. Flaw: In our pilot study, we showed sneaky samples to the trainers. This study has a flaw where samples from earlier rounds are shown first to the trainers. The only plot in the main text that came from the pilot study is Figure 7b (human judge accuracy of sneaky samples across rounds); two more in the appendix for ablations (S7c, S8). Most of the human eval results came from the main study with larger sample size and fully randomized assignment (75 trainers, 15983 samples). Trainers are randomly assigned balanced correct / incorrect samples from the last helpful checkpoint each round. Most of the human eval results (Fig 1, 4, 8) are from the main study, so our conclusions on poor legibility of correctness-only baseline, increased legibility on helpful samples, and decreased time to solve are all accurate. We didn’t re-run human eval on sneaky solutions and ablations due to budget constraints.
>
> 4. Distribution of problems: The problems are uniformly random GSM test problems, and no one sees any single problem more than once. We expect some learning and fatigue effects so we limit total labor to 4-hour max in the main study. Since assignment is fully randomized this should not affect the relationship between judge accuracy and training rounds. We refer the reviewer to Appendix K for full details.
>
> Single dataset and generalizability: Most previous works on scalable oversight only study the QuaLITY dataset [Irving et al., 2018; Radhakrishnan, 2023; Michael et al., 2023; Khan et al., 2024]. Our work is the first to study debate-like methods between provers and verifiers where the gap isn’t privileged information but capability, and on a reasoning domain. GSM8k is a gateway reasoning dataset where there is a sizable capability gap between current large vs small LLMs. We agree that extensions to diverse datasets are important future directions. We’re excited to address this in future work.
>
> Definition of legibility: In section 1,Line 33-34, we defined legible explanations as ‘outputs that can be fully understood and verified by humans or other trusted systems.’ We further defined checkability as soundness and completeness (Line 46-49) and operationalized mathematically in section 3. To measure human legibility, we used time-constrained judge accuracy, which is directly and holistically measuring a human's ability to check the correctness of LLM outputs, based on our definition. Decomposing human legibility as brevity, format, easy vocabulary, etc., is a good future direction for human-computer interaction study. We are excited to hear any constructive suggestions for better definitions.
>
> To recap, our main contributions are articulating and formulating an important new problem in alignment, as well as coming up with ways to measure and mitigate performance-legibility tradeoff in a nontrivial reasoning domain. It is truly surprising that checkability by smaller LLMs transfers to human legibility.
>
> We’ve updated our submission pdf thanks to the reviewer’s helpful feedback.

---

> > ### Author Response · Authors · 2024-11-23
> >
> > Dear reviewer 5yX2, thanks again for your helpful feedback! We’ve addressed your concerns in the response above and would greatly appreciate it if you could revisit your evaluation. Let us know if you have any further questions!

---

> ### Comment · Reviewer_5yX2 · 2024-11-24
>
> Thank you for your responses. I have read through them, and also the other reviews and associated responses.
>
> I now have greater concerns regarding the weaknesses of this paper, and have the view that this paper does not meet the bar for an ICLR publications.
> - As the authors have emphasized, they believe that their main contributions are focused around formulating a new problem and proposing the notion of legibility. However, the issue I have with this is that the notion of legibility is not analyzed or characterized sufficiently for this to be a major contribution on its own. The assertion is that legibility is best analyzed using short-time trials and accuracy, but no detailed ablation or analysis could be done on this regarding confounding factors or other contributors, nor analysis of alternative metrics, which is what I would expect in a paper focused on this issue. Several past papers have already considered the importance of humans being able to easily read (e.g. readability vs legibility) or verify the output of the LLMs for practical deployment of LLMs, such as [1]. The novelty of this work with respect to other such works, including explainability works that do not emphasis solely on faithfulness but more general forms of interpretability and readability, appears limited in light of the little details on this regard.
> - The paper is also largely empirical in supporting its key contributions, but unfortunately do not meet the rigor expected of primarily empirical work, as also pointed out by other reviewers though they have provided higher scores despite the very preliminary work due to novelty (which I have raised concerns on above). It is not possible to assess whether the results are generalizable beyond this specific dataset or LLM model, especially when the training process involves many hyper parameters and can be aggressively fine-tuned for producing good results -- all points raised by other reviewers as well. The authors' assertion regarding their belief that the results would be generalizable to other models or simple datasets are unsubstantiated. There are also still figures in the main paper that are based on flawed experimental set-ups, which though explicitly mentioned in the paper, still do not warrant them being placed and discussed in the main paper. The point that most previous works on debate-like methods only studied one dataset is also over-claimed as there are several multi-agent debate works (e.g. [2]) that focuses on reasoning tasks (multiple datasets) though not focused on the same direction of 'legibility'. Nevertheless, the deficiency of past works should never be justification for that of a work under review, unless there are real limitations in the practicality or availability of other relevant models or datasets, which is not a case now.
>
> My suggestion is that the authors do some additional experiments on another dataset and other models, even if smaller ones, to at least provide some hints of basic generalizability beyond the single set-up that have been presented in this current version of the work. If the authors would like to primarily claim the novelty of 'legibility' and the associated research direction, I suggest that the authors flesh out the framework and analysis underlying this research direction, show more detailed ablation studies on important related factors and alternative metrics, articulate clearly the gaps compared to previous related works especially related to explainability/interpretability beyond faithfulness, readability in past works, and actor-critic frameworks, and present clear paradigm changes along with future directions and associated impact.
>
> Given the above, I believe that the work, while promising, is still too preliminary to meet the bar for an ICLR publication. Hence, I lower my score.
>
> [1] Accuracy, readability, and understandability of large language models for prostate cancer information to the public. 2024
> [2] Improving Factuality and Reasoning in Language Models through Multiagent Debate, 2023

---

> > ### Comment · Reviewer_kzyP · 2024-11-28
> >
> > After considering this review, I've reduced my confidence level, though my score remains unchanged.
> >
> > The first reason is that I overlooked the complexities involved in defining legibility, and the paper's approach to this indeed seems somewhat arbitrary. I'm maintaining my score as I still believe this work meets the ICLR acceptance threshold, even if the specific legibility metric used may measure something slightly different and might not transfer. The readability reference in the medical domain (or other non-math domains) seem less relevant here.
> >
> > The second issue still remains. I'd appreciate if the authors could elaborate on the novelty of the direction of this paper in the debate domain, particularly in relation to the "Improving Factuality and Reasoning in Language Models through Multiagent Debate" work from 2023 mentioned in the comment above. The "Scalable oversight" paragraph in the Related Work section doesn't address this, and I believe it warrants further discussion.

---

> > ### Author Response · Authors · 2024-12-01
> > **Clarification of our conceptual and empirical contributions (Part 1/2)**
> >
> > We thank the reviewer for clarifying their two main concerns:
> >
> > 1. Novelty of legibility problem formulation and measurement of legibility:
> >
> > Explainability literature has mostly been focused on uncovering how AI models arrive at a prediction — “Explainable AI (XAI) refers to techniques that provide human-understandable insights into the workings of AI models.” [Wu et al] Prominent works in LLM explainability include autoencoders [Gao et al], circuit discovery [Conmy et al], automatically explain neuron-activations [Bills et al]. The legibility problem we study is distinctively different from those works because the ‘explanation’ the model provides via CoT can be entirely different from how models arrive at the prediction. It is not our goal to unbox the model’s workings in any way.
> >
> > Our paper is a scalable oversight [Bowman et al] paper that studies how to make a more capable system trustworthy by another less capable system. Solving the scalable oversight problem doesn’t necessarily solve explainability, or vice versa.
> >
> > There are papers focused on readability and legibility of AI’s outputs in specific application domains. We have cited [Dragan et al; Oliveira et al; Wen et al] in robotics and coding domains (Line 158-159). The prostate cancer paper mentioned by the reviewer is more applied, leaning towards human-computer interaction research. The application papers each studied a specific instantiation of the legibility problem without considering it more generally as a single ML problem. Hence our main conceptual contribution is defining the problem mathematically, making it amenable to ML study both theoretically and empirically. It provides a useful framework to analyze the application papers — When the provers (e.g. medical information systems) and the verifiers (e.g. human patients) have certain capability gaps, how do we finetune the systems to be more legible to the patients? With more general ML techniques like PVG, we can improve legibility not just for medical information, but also for all other applications like robotics [Dragan et al] and coding [Wen et al]. Therefore, we think our problem formulation is one major conceptual contribution.
> >
> > On the choice of using time-limited judge accuracy as measure of legibility: Prior work like [Parrish et al] also used this. We acknowledge that this is only one of many possible measures. The reason we picked this is because we are mainly concerned about whether we can build trust in a system more capable than us in the future, and the ability to spot its mistakes is the crux of building trust. Time limit is natural because humans do not have infinite time to check LLM output (think AI medical coding). We fully admit that the 45s limit in our experiment is because the GSM dataset is easy; given more time humans would make fewer mistakes. This is meant to simulate human evaluation of more difficult problems (where we would have a longer limit) without having to train on difficult datasets that require more compute and larger LLMs.
> >
> > In short, time-limited judge accuracy is not the only or the best legibility measure, but it is a valid one, has precedent in prior literature, and is well-motivated by the problem (checkability).

---

> > > ### Author Response · Authors · 2024-12-01
> > > **Clarification of our conceptual and empirical contributions (Part 2/2)**
> > >
> > > 2. On empirical contributions:
> > >
> > > We thank the reviewer for bringing [Du et al] into our attention. The main focus of [Du et al.], as well as other multiagent works on reasoning datasets [Liang et al; Wang et al], is to scale test-time inference of LLMs, so after some deliberation they achieve higher accuracy. The core mission of those papers is to increase the final performance, with no benefit to the scalable oversight problem, because there is no “prover” vs. “verifier” in their problem definition. The main problem we tackle is how to make a less capable system able to trust a more capable system, when the less capable system cannot solve the problems on its own in a certain reasoning domain. This is a problem both for larger LLMs vs. humans, and for larger LLMs vs. smaller LLMs, since we expect next-generation LLMs to outperform humans on many tasks soon, and also we want to use smaller LLMs to aid us in calibrating trust because less capable LLMs is more unlikely to be deceptively aligned [Hagendorf et al; O’Gara et al].
> > >
> > > Hence our work is more comparable to similar scalable oversight works like [Irving et al; Parrish et al; Michael et al] etc. already cited in Line 116-130, which study the type of debate where a less capable system (human or LLM) is the judge and the more capable system (LLM) is the prover.
> > >
> > > We fully acknowledge the deficiency of only experimenting on the GSM dataset, and would like to experiment on other reasoning databases such as those in [Du et al] in future work. We will also cite those works in the next revision due to similarity in debate-like methods. Nevertheless, we maintain that no prior work in the literature has empirically measured and addressed the legibility gap in a reasoning domain at scale, specifically by continuously varying the capability gap between the prover and the verifier and quantitatively assessing the trade-off between prover performance and checkability. This represents the primary empirical contribution of our work.
> > >
> > > [1] Wu et al. Usable XAI: 10 Strategies Towards Exploiting Explainability in the LLM Era.
> > >
> > > [2] Gao et al. Scaling and evaluating sparse autoencoders.
> > >
> > > [3] Conmy et al. Towards Automated Circuit Discovery for Mechanistic Interpretability.
> > >
> > > [4] Bills et al. Language models can explain neurons in language models.
> > >
> > > [5] Bowman et al. Measuring progress on scalable oversight for large language models.
> > >
> > > [6] Parrish et al. Two-Turn Debate Does Not Help Humans Answer Hard Reading Comprehension Questions.
> > >
> > > [7] Liang et al. Encouraging Divergent Thinking in Large Language Models through Multi-Agent Debate.
> > >
> > > [8] Wang et al. Self-Consistency Boosts Calibration for Math Reasoning.
> > >
> > > [9] Hagendorf et al. Deception abilities emerged in large language models.
> > >
> > > [10] O’Gara et al. Hoodwinked: Deception and cooperation in a text-based game for language models.

---

### Official Review · Reviewer_PYjp · 2024-11-04

**Soundness:** 2
**Presentation:** 2
**Contribution:** 2
**Rating:** 5
**Confidence:** 3

**Summary:**

This paper investigates how to make outputs from large language models (LLMs) more legible and reliable through a method inspired by Prover-Verifier Games (PVG). The authors focus on the challenge of maintaining both correctness and legibility in outputs when solving grade-school math problems. The proposed iterative training algorithm trains a verifier to predict solution correctness and conditions, and provers to create either correct or subtly incorrect solutions. This approach enhances verifier robustness and human assessability of solutions over time.

**Strengths:**

- The paper presents an innovative adaptation of the Prover-Verifier Game to train LLMs for legibility.
- It includes both theoretical proofs and empirical studies showing the benefits of their method in improving solution checkability.
- The study extends beyond automated verification to demonstrate human evaluators' performance, indicating real-world applicability.
- The authors acknowledge trade-offs between optimizing for accuracy and maintaining human-legible outputs, highlighting practical insights for future applications.

**Weaknesses:**

- The study primarily focuses on grade-school math problems; exploring broader applications could demonstrate the method's generalizability.
- The paper could benefit from more discussion on integrating this training into existing LLM frameworks and the computational resources required.

**Questions:**

- Could the proposed prover-verifier training framework be adapted for complex, non-mathematical reasoning tasks such as legal or medical document analysis?
- What are the practical challenges in deploying this approach for real-time systems where rapid responses are required?
- How does the model handle ambiguous or contextually complex questions where the verifier might struggle to discern correctness?
- Would integrating human-in-the-loop feedback during training rounds enhance the verifiers' robustness and prover legibility further?

---

> ### Author Response · Authors · 2024-11-17
>
> We thank the reviewer for recognizing our empirical and theoretical contributions as well as real-world applicability based on evidence from human evaluations.
>
> Single dataset and generalizability: Most previous works on scalable oversight only study the QuaLITY dataset [Irving et al., 2018; Radhakrishnan, 2023; Michael et al., 2023; Khan et al., 2024]. Our work is the first to study debate-like methods between provers and verifiers where the gap isn’t privileged information but capability, and on a reasoning domain. GSM8k is a gateway reasoning dataset where there is a sizable capability gap between current large vs small LLMs. It is relatively toy, and we are excited to extend the checkability training framework to harder math datasets and coding, as well as fuzzier domains such as human preference data (Ouyang et al., 2022) in future work. Our method can readily be applied to non-mathematical domains (including legal and medical), as long as we have a subset of problems labeled with ground truth answers to calibrate the verifier’s confidence in the prover's solutions. For example, for the MedQA dataset [Jin et al. 2020], we can train a prover (larger LLM) to output a multiple choice answer along with a CoT justification, and the verifier to classify whether the given solution is correct. If there’s a capability gap between the two models, the verifier will need to use the justifications, and hence PVG training will encourage the prover to generate more checkable justifications and the verifier to discern good from bad solutions.
>
> Incorporation into current LLM frameworks: Our experiments are already on current LLMs (GPT-4 series). We’re excited to try harder datasets at an even larger scale, but have to leave them for future work. The reviewer has concerns about real-time systems where rapid responses are required. We clarify that checkability training is a train-time procedure. At inference time the helpful prover will output a CoT solution with the answer, which should be more checkable by smaller LLMs / humans than baseline. With high-stake scenarios (e.g. writing GPU kernel code) humans will check the code before deploying it; with low-stake scenarios we can just use the final answer and the additional justification increases the answer's legitimacy.
>
> Computational resources used in this paper: 6 rounds of PVG training takes 2400 A100-40G GPU hours (we updated our paper Line 531-532).
>
> Contextually complex questions: Our method is fully general as long as there is a finite set of problems with ground truth answers. If an objective ground truth answer exists, even if they’re long and complex such as human preference data (Ouyang et al., 2022), we can use a grader (another LLM) that compares the policy’s answer with the ground truth answer and gives reward when they’re semantically similar. When we do not have ground truth answers, we are in the ‘unsupervised’ setup discussed in Section 5.1 and future work is needed to develop new algorithms such as self-consistency [Wang et al 2023]. Our method is fully general, but reasoning-heavy tasks will probably benefit the most. We mainly target high-stake domains that require natural language justification to build trust in LLM outputs, such as math, coding, legal, or medicine.
>
> Integrating human-in-the-loop feedback can certainly enhance checkability training. We aimed to develop PVG as an alternative to RLHF, but a hybrid is also conceivable. We can incorporate signals from which solutions trainers got right / wrong into training. We can possibly finetune the helpful policy to clone solutions trainers got right more frequently. This is an exciting future direction.
>
> We’ve updated the submission based on the reviewer’s feedback on clarification on applicability and computational resources. Please review Section 5.1 and Appendix B for more details.

---

> > ### Author Response · Authors · 2024-11-23
> >
> > Dear reviewer PYjp, thanks again for your helpful feedback! We’ve addressed your concerns in the response above and would greatly appreciate it if you could revisit your evaluation. Let us know if you have any further questions!

---

> > > ### Comment · Area_Chair_TGUU · 2024-11-26
> > >
> > > Dear reviewer PYjp,
> > >
> > > Could you please respond to authors' rebuttal and see if you would like to update your review? Thanks very much!
> > >
> > > AC

---

### Official Review · Reviewer_kzyP · 2024-11-04

**Soundness:** 3
**Presentation:** 4
**Contribution:** 3
**Rating:** 8
**Confidence:** 2

**Summary:**

This paper investigates the trade-off between performance and legibility in large language models (LLMs) solving  GSM8k augmented with 100k synthetic examples.
The authors find, via a human study, that RL training for correctness reduces legibility of the solutions.
To improve legibility, the papers propose a "checkability training" method, similar to “Learning to Give Checkable Answers with Prover-Verifier Games” (Anil et al., 2021), which is in turn inspired by prior research in interactive proofs and PAC learning.
This involves iterative training of small verifier models and larger "helpful" and "sneaky" prover models.
Checkability training achieves a balance, resulting in more legible solutions at a modest cost to accuracy.

**Strengths:**

I find the experiments very well-motivated.   Prior papers in adjacent areas like debate focus on question-answering datasets;
but it is obvious that research on legibility of reasoning is much more important.

Although the GSM dataset is easy, this is the first paper in this direction, and honestly the experiments would have been a good contribution even if it was on a toy dataset. I also think human legibility studies are less likely to be misleading on this sort of dataset.

The paper is extremely well-written, and also proposes many interesting follow-up ideas. I am particularly interested in whether sneaky provers resulting from this sort of training are useful in other ways, for example, as model organisms for deception evals.

**Weaknesses:**

**Models:** My assessment of the paper is based on the assumption that it does not matter for the purpose of this conference that the models here never available to the public in any form.


In the interest of taking everything in good faith, I see two acceptable reasons for this:
-   there is no herd of similar models over a range of compute scales used in the paper; or
-   human studies had to start before models of similar capabilities were available to the public;

This also assumes that the models used in the paper are similar in all important ways to other LLMs that the research field is aware of. I am open to reducing my score to 6 if there is a reviewer consensus that there is no good reason to do this research on the models used in the paper.


**Ground truth answers:**
As mentioned in the paper, the rewards in checkability training require ground truth labels. Most of the important applications of this line of research will be in settings where there are no ground truth labels. Given that, I am uncertain whether the method in this paper will play a major role in long-term scalable oversight research.


**Technical details:** I believe the training described in the paper might be tricky to reproduce on a different suite of models, due to the lack of details about optimization, learning rates, and so on. Would the authors be interested in disclosing whether this sort of iterative training is difficult to set up correctly, or it worked on basicaly the first try?

**Questions:**

Claim about prior work: Figure 4 shows a drop in legibility due to optimization for correctness. Are the authors sure this phenomenon is novel? I cannot find anything right now, but I personally thought this was the case and already established somewhere.

---

> ### Author Response · Authors · 2024-11-17
>
> We thank the reviewer for recognizing that our experiments are well-motivated, we made a first step on the reasoning domain, and the paper is well-written.
>
> Availability of models: Although our main setup has a prover-verifier size gaps of 10^-3, OOD evals in Figure 6 and ablation studies in Appendix A.4, F, G require varying the size gap continuously. Using models of different sizes (10^{-4, -3, -2, -1, 0}) from the same series allows for study of a fine-grained scaling law between prover-verifier capability gap and legibility, holding other factors (pertaining datamix, algorithms, etc.) constant. Open-source models (e.g. LLAMA 8B vs 70B) would yield fewer, isolated data points. We do believe that we would obtain qualitatively the same results with other autoregressive LLMs.
>
> Supervised learning: We see PVG being useful as long as we have a small set of labeled data to calibrate the verifier’s confidence, and the prover can be trained without labeled data. For example, in the iterative goodharting training setup (Appendix A.4), the policy solely maximizes the verifier score, and the verifier is trained on a fixed set of problems with ground truth labels but using rollouts from an increasingly diverse set of policies. Hence this can potentially be a data-efficient semi-supervised setup. Additionally, we can potentially leverage self-consistency to train the verifier unsupervised (e.g. it cannot accept two solutions leading to different answers). Additionally, we hope to leverage domain generalization so after the verifier has been trained on several labeled domains it can generalize to novel unlabeled domains (Section 5.1, Line 488-493). All of these are meaningful future directions.
>
> Technical details: The main ablation study we performed include the reward function (Appendix A), verifier size (Appendix F), and initialization (Appendix H). We found that the reward function matters; verifier size within a certain range does not matter but too large / small does not work; the initialization does not matter. Additionally, we found no qualitative difference between early stopping vs always training each round to 8k steps (Section 5.1, Line 522-523). We used the standard RLHF PPO hyperparameters including optimizer, learning rate, and KL regularization coefficient and did not tune at all for this paper. Overall the training dynamics in Figure 5 worked on the first try as long as we used the Signed Relative Convincingness (SRC) reward. 6 rounds of PVG training takes 2400 A100-40G GPU hours (updated draft Line 531-532).
>
> Prior work: The ‘performance tax’ has been discussed conceptually in Distinguishing three alignment taxes [Leike, 2022]. Empirically several works have found and tried to mitigate performance degradation during instruction-following training [Ouyang et al. 2022, Glaese et al. 2022, Bai et al. 2022]. ‘Legibility tax’ as a type of performance tax has been less explored.
>
> We’ve updated our submission pdf thanks to the reviewer’s helpful feedback.

---

> > ### Comment · Reviewer_kzyP · 2024-11-23
> >
> > Thank you for the reply! My concerns about the models and the training setup generalizing to other models have been partly addressed; as other reviewers did not raise this concern, I keep my score.
> >
> > I have also read the other reviews, and agree with the authors that the GSM dataset (or something similarly distributed) is a good enough test bed for the first paper in this research direction.

---

### Official Review · Reviewer_qnLm · 2024-11-09

**Soundness:** 3
**Presentation:** 2
**Contribution:** 4
**Rating:** 8
**Confidence:** 4

**Summary:**

The paper presents work on an important direction in AI Alignment - making the outputs of highly capable LLMs legible to humans. It describes results on a setup consisting of iterative prover-verifier training which improves accuracy and legibility. It also analyzes training dynamics (e.g. impact of the verifier’s size on iterative training), rendering the paper overall quite impactful in terms of building systems that have outputs legible to human overseers.

**Strengths:**

- Overall, this is a strong paper presenting alignment research in a novel, viable, and important direction. It focuses explicitly on training setups that are understudied and demonstrates strong results around legibility.
- The paper rests on a strong theoretical foundation around prover-verifier games, takes into account how an adversarial prover might work, and presents important early results about points such as verifier sizes, iterative setups when it comes to training models towards legibility/alignment, etc.
- The paper addresses concerns such as reward hacking and provides a good amount of diversity around reward functions and how these could influence future research.
- The paper is open about its limitations and future work around unsupervised learning for tasks that lack ground truth labels.

**Weaknesses:**

- The paper conducts all experiments exclusively on GSM8k, where explanations can indeed be step by step while being natural. Moreover, all experiments are done on a single model type (GPT-4). This raises some concerns about the generalizability of the prover-verifier setup, especially to domains such as code generation, writing, etc. and settings with different base models.
- The iterative training process might lead to overfitting and the early stopping conditions don’t seem clear and generalizable. The prover and verifier could adapt to each other’s outputs, and given that there is no cross model testing, it’s difficult to make claims about generalization in this setting.
- Lack of experiments/comparison with existing work on Explainable AI. Work on multi-agent debate for safety/legibility [1] or other methods to improve legibility [2] are not compared against.
- The iterative training setup introduces risks such as reward hacking, deception, collusion, or steganography. The paper does not address concerns highlighted in alignment literature such as [3][4][5][6][7] that this work directly impacts. The verifier could be jailbroken by a strong prover with steganographic methods arising out of training, or models may converge to deceptive strategies that jointly lead to higher rewards. It would be important to discuss the paper’s results against such prior work in safety.
- There are some points which I have raised below in my questions, such as details on the flawed campaign for collecting human data or the accuracy of synthetic GSM8k data or early stopping conditions/hyperparameter tuning.


[1]Debating with More Persuasive LLMs Leads to More Truthful Answers

[2] Usable XAI: 10 Strategies Towards Exploiting Explainability in the LLM Era

[3] Deceptive Alignment Monitoring

[4] Preventing Language Models From Hiding Their Reasoning

[5] Secret Collusion Among Generative AI Agents

[6] AI Control: Improving Safety Despite Intentional Subversion

[7] AI deception: A survey of examples, risks, and potential solutions

**Questions:**

- How are the synthetic data samples for GSM8k checked for correctness? Are they checked for correctness?
- Could the authors provide more details on the ‘flawed campaign’ regarding human annotations in their experiments?
- Risk of overfitting the prover and verifier to each other’s outputs: what happens if one uses different models? How does this then extend to other domains? Will the same amount of training resources be required for legibility in every domain or are there any generalization results that the authors could present?
- What issues related to the safety literature highlighted in the weaknesses section do the authors see arising out of their iterative training setup, and how can early work be done to prevent such alignment issues?

---

> ### Author Response · Authors · 2024-11-17
> **Response to weaknesses**
>
> We thank the reviewer for recognizing that we formulated an important alignment problem in a concrete ML setting and made novel contributions both theoretically and empirically.
>
> Generalizability:
> Most previous works on debates only study the QuaLITY dataset [Irving et al., 2018; Radhakrishnan, 2023; Michael et al., 2023; Khan et al., 2024] where the prover has privileged information. Our work is the first to study debate-like methods between provers and verifiers where the gap isn’t privileged information but capability, and on a reasoning domain. GSM8k is a gateway reasoning dataset where there is a sizable capability gap between current large vs small LLMs.
> Using models of different sizes (10^{-4, -3, -2, -1, 0}) from the same series allows for study of a fine-grained scaling law between prover-verifier capability gap and legibility, holding other factors (pertaining datamix, algorithms, etc.) constant. Open-source models (e.g. LLAMA 8B vs 70B) would yield fewer, isolated data points. We do believe that we would obtain qualitatively the same results with other autoregressive LLMs.
> We agree that extensions to diverse datasets and model families are important future directions. We hope to have made a first step inviting further works on measuring and mitigating legibility gaps across different setups.
>
> Overfitting:
> 	We stop each iteration either when the sneaky ‘wins’, e.g. sneaky incorrect sample becomes as convincing as helpful correct, or until 8k RL steps. The former is due to reaching an equilibrium (the sneaky will always win from that point on), and the latter is due to compute constraints. We observe no qualitative difference in training dynamics between early-stopping or non early-stopping setups - they are robust to early-stopping, initialization (Fig S11) as well as hyperparameters (Section 5.1).
>
> Co-adaptation:
> 	We clarify that the goal of checkability training is to make the verifier recognize a particular helpful policy’s correct answers as correct, while being robust to wrong answers from adversarial sneaky policies. It is expected that the verifier prefers the helpful prover’s completions over an off-policy correct sample, because completeness property only requires that there exists one helpful prover, not that the verifier has to accept all correct samples. Hence our verifier is expected to co-adapt with the helpful policy, but should also be robust to novel adversarial sneaky policies. Our existing robustness study is in Section 4.3.1, where we demonstrated that the verifier is becoming more robust to at least two types of unseen attacks (adversaries optimizing CGC and goodharting reward) as well as more robust on best-of-n eval.
> 	We did train the verifier and the policies in a coordinate-descent fashion, but coordinate descent isn’t equivalent to collusion. As in most multiplayer games, we do not expect to find a global but rather local equilibrium. In each iteration, the prover is expected to try new ways to fool the verifier, and the verifier is expected to learn from its past mistakes. What we found is that after several iterations, it got harder and harder for the sneaky prover to fool the verifier, as the verifier has been trained on more and more types of hacks. We do not claim that the verifier has achieved perfect soundness, since this is a local, not global equilibrium. We showed evidence that it costs increasing compute to find a hack (Fig 6).
>
> Related literature on debate and explainability:
> 	We have updated the related works section. We thank the reviewer for bringing up those works. Explainability focuses on revealing how the model derived at the output whereas legibility focuses on producing a good proof of the final answer independent of how the model derived at the answer.
>
> Relations to other alignment problems:
> 	We study a new alignment problem (legibility) that is quite different from the other problems mentioned by the reviewer, and it’s not clear why checkability training will introduce those risks. PVG could be a way to reduce reward hacking [Gao et al., 2023; Skalse et al., 2022] and we refer the reviewer to Appendix A.4 for demonstration. Similarly, if the model produces more human-legible outputs (as we showed in human evals) this reduces the risk of deceptive alignment [Park et al. 2024] and increases ease of monitoring [Carranza et al.]. The concern that our provers and verifiers may collude and develop secret stenographic protocols illegible to humans is valid, but we didn’t observe that for GSM dataset based on human eval. It’s true that the solutions verifiable by smaller LLMs can possibly be non-human-legible, so we think transferability to human legibility is an interesting finding of our paper.

---

> ### Author Response · Authors · 2024-11-17
> **Answers to questions**
>
> 1. Human spot-checking showed that the synthetic samples are 95% correct. We trained models on the same amount of real vs. synthetic samples and the (real) test set performance is similar.
> 2. Details about the human campaign are in Appendix K. Fig. 7b, S7c and S8 are affected since samples from earlier rounds are shown first to trainers. All other human eval results (Fig 1, 4, 8) are unaffected.
> 3. It is an interesting future direction to see whether performing checkability training on a few domains can transfer to other domains. We are excited to try this out in future work.
> 4. Promising future directions are discussed in 5.1 and Appendix B. We suggest giving the prover a scratchpad space for CoT and let it decide what to show to the verifier. This is an easy improvement to avoid exacerbating deceptive alignment.
>
> We’ve updated our submission pdf thanks to the reviewer’s helpful feedback.

---

> > ### Comment · Reviewer_qnLm · 2024-11-23
> >
> > Thank you to the authors for responding. Most of my concerns have been addressed and I will raise my score. I would like to highlight that my updated score is based on the fact that this work is pursuing a novel direction. It is, however, quite preliminary and I would strongly recommend the authors to spend a Section in the Appendix discussing very concrete directions for the community based on this research.

---

### Meta-Review · Area_Chair_TGUU · 2024-12-24

**Metareview:**

This paper presented a new direction "legibility", clear and easy to check as one of the alignment goals. The authors proposed
a training algorithm inspired by Prover-Verifier Game by training small verifiers to improve the legibility of the LLM outputs.

Strength:
1. An potentially very important alignment aspect: legibility.
2. A novel method to improve the legibility

Weakness:
Though the work is very interesting, it is a bit preliminary in terms of 1) experimental validation is mainly on GSM8k. So the generalization of the method is difficult to known. The authors provided some argument along this line, though it doesn't seem to be sufficient to address the concern. 2) The definition of `legibility` is not very formal and not adequately ablated against other confounding factors, mostly argued by reviewer 5yX2.

Overall, I believe this could be an important paper, though the current form is a bit preliminary and I would encourage the authors to take into the comments to make this paper stronger.

**Additional Comments On Reviewer Discussion:**

Reviewers actively participated in the discussion during rebuttal periods. The paper got diverse scores from 3, 5, 8, 8. The reviewers with higher scores were mostly on the novelty and also have concerns about the limitations of experimentations and results are preliminary.

---

### Decision · Program_Chairs · 2025-01-22

Reject